# Nutritional Profile and Health Benefits of *Ganoderma lucidum* “Lingzhi, Reishi, or Mannentake” as Functional Foods: Current Scenario and Future Perspectives

**DOI:** 10.3390/foods11071030

**Published:** 2022-04-01

**Authors:** Aly Farag El Sheikha

**Affiliations:** 1College of Bioscience and Bioengineering, Jiangxi Agricultural University, 1101 Zhimin Road, Nanchang 330045, China; elsheikha_aly@yahoo.com; 2Department of Biology, McMaster University, 1280 Main St. West, Hamilton, ON L8S 4K1, Canada; 3School of Nutrition Sciences, Faculty of Health Sciences, University of Ottawa, 25 University Private, Ottawa, ON K1N 6N5, Canada; 4Bioengineering and Technological Research Centre for Edible and Medicinal Fungi, Jiangxi Agricultural University, 1101 Zhimin Road, Nanchang 330045, China; 5Jiangxi Key Laboratory for Conservation and Utilization of Fungal Resources, Jiangxi Agricultural University, 1101 Zhimin Road, Nanchang 330045, China; 6Department of Food Science and Technology, Faculty of Agriculture, Minufiya University, Shibin El Kom 32511, Egypt

**Keywords:** *Ganoderma lucidum*, prebiotics, functional food, therapeutic properties, antiviral drugs, COVID-19, health risks, geo-tracing

## Abstract

*Ganoderma lucidum* has a long history of medicinal uses in the Far East countries of more than 2000 years due to its healing properties. Recently, *G*. *lucidum* has come under scientific scrutiny to evaluate its content of bioactive components that affect human physiology, and has been exploited for potent components in the pharmacology, nutraceuticals, and cosmetics industries. For instance, evidence is accumulating on the potential of this mushroom species as a promising antiviral medicine for treating many viral diseases, such as dengue virus, enterovirus 71, and recently coronavirus disease of 2019 (COVID-19). Still, more research studies on the biotherapeutic components of *G*. *lucidum* are needed to ensure the safety and efficiency of *G*. *lucidum* and promote the development of commercial functional foods. This paper provides an extensive overview of the nutraceutical value of *Ganoderma lucidum* and the development of commercial functional food. Moreover, the geo-origin tracing strategies of this mushroom and its products are discussed, a highly important parameter to ensure product quality and safety. The discussed features will open new avenues and reveal more secrets to widely utilizing this mushroom in many industrial fields; i.e., pharmaceutical and nutritional ones, which will positively reflect the global economy.

## 1. Introduction

### 1.1. What Does History Say about Ganoderma lucidum?

“Lingzhi is a miraculous king of herbs”—Chinese people (221–206 BC).

Historically, the Romans considered mushrooms in general as the food of their gods and only served them for great feasts, while the Greeks and the Vikings believed that eating mushrooms gave them strength and enthusiasm before the war. America’s indigenous people have often used mushrooms in age-old rituals (e.g., magical hallucinogens) to cross the body and mental barrier [1]. Considered as one of the main folk medicinal mushrooms, *G*. *lucidum* was used for many centuries and reported under several names in China (Lingzhi), Japan (Reishi), and Korea (Mannentake). According to bimillennial beliefs, *G*. *lucidum* can promote health and longevity, but it was also considered a combination of spiritual force and a source of immortality [2,3,4]. Moreover, the Japanese people have regarded this mushroom as a “10,000-year” mushroom [5,6,7].

Several researchers have pointed out the long history of traditional medicinal uses of mushrooms, especially *G*. *lucidum*, mostly in Far East countries, dating back more than 4000 years [7,8,9,10,11,12,13,14]. This type of mushroom has therapeutic characteristics with medical claims that can be attributed to a well-respected pharmacopeia from the Qin dynasty (221–206 BC) called *Shen Nong Ben Cao Jing,* or *The Divine Farmer’s Materia Medica* [13,15]. The ethnomedicinal uses of *G*. *lucidum* had reflections on culture, such as the artworks beginning in the Yuan Dynasty (1280–1368 AD) [7,13]. This was not limited to artworks, but the use of *G*. *lucidum* images extended to furniture, carvings, paintings, and even women’s accessories [2].

For a long time, *G*. *lucidum* has been used as a traditional medicine for treating neurasthenia, debility of prolonged illness, insomnia, anorexia, dizziness, chronic hepatitis, hypercholesterolemia, mushroom poisoning, coronary heart disease, hypertension, prevention of acute mountain sickness, “deficiency fatigue”, carcinoma, and bronchial cough in the elderly [16,17]. Studies on medicinal mushrooms began in Western science more than 30 years ago. These studies have continued until the present via a series of exciting discoveries related to the biological activities of *Ganoderma lucidum*, including antitumor and anti-inflammatory effects, as well as cytotoxicity to hepatoma cells [18,19].

### 1.2. Ganoderma lucidum through the Glasses of Botanists, Taxonomists, Economists, and Scientometric Analysis

#### 1.2.1. Through Botanists’ Glasses

Morphologically, *lucidum* is a word derived from the Latin word *lucidus,* which means “shiny” or “brilliant”, and describes the varnished look of the mushroom’s surface. Overall, *G*. *lucidum* is a large, dark mushroom distinctively characterized by a glossy surface (including a red-varnished and kidney-shaped cap) and a woody texture (see Figure 1). The fresh mushroom is soft, corklike, flat, lacks gills on its underside, and releases its spores via fine pores. The pore color on its underside depends on the age of the mushroom, and maybe white or brown [6,20]. Chen [21] described the nature of *G*. *lucidum*’s growth on the bases and stumps of a wide variety of deciduous trees, such as oak, maple, elm, willow, sweetgum, magnolia, and locust, and less frequently found on coniferous trees (e.g., larix, ptea, pinus) in Europe, Asia, and North and South America, especially in temperate rather than subtropical regions.

#### 1.2.2. Through Taxonomists’ Glasses

*Ganoderma lucidum* (Curtis) P.Karst. was first described by Curtis [22] based on material from England, and the description was sanctioned by Fries [23]. The first scientific record of *G*. *lucidum* from China was made by Teng [24] when he incorrectly identified a Lingzhi specimen as *G*. *lucidum*. Geographically, the *G*. *lucidum* sensu stricto (Curtis) Karst mushroom is native to Europe and some parts of China [25]. According to the Index Fungorum (2016) (http://www.indexfungorum.org, accessed on 16 February 2022), *Ganoderma lucidum* (Curt: Fr.) Karst. belongs to Basidiomycota (phylum), Polyporales (order), and Ganodermataceae (family), as classified by the taxonomist Nahata [5]:Kingdom: FungiDivision: BasidiomycotaClass: AgaricomycetesOrder: PolyporalesFamily: GanodermataceaeGenus: GanodermaSpecies: *G*. *lucidum*

#### 1.2.3. Through Economists’ Glasses

*Ganoderma*-based products attract a great deal of interest in many countries within Europe and North America, although South Asia (Malaysia, Singapore, China, Japan, and Korea) are the principal producers/providers of these food products [26]. In the past, consumption of *G*. *lucidum* was restricted to the wealthy only, and therefore there was no need to expand its cultivation, and what was grown in the wild was sufficient. Recently, however, the consumption of this mushroom has increased through multiple societal groups as an effective alternative to modern medicine or alongside it, and this is what has called for the expansion of its cultivation [27,28]. With over 110,000 ton/year, China is the biggest producer and exporter of *G*. *lucidum* [29]. Therefore, *G*. *lucidum*-based products play a pivotal role in the Chinese economy as a source of foreign-exchange flow through increasing exports and as promising products at the food and medical levels.

Generally, the mushroom’s ingredients possess a wide variety of biological properties, including pharmaceutical, nutraceutical, and cosmetic, as shown in Figure 2 [8,30,31]. As such, regarding the *G*. *lucidum* mushroom, there are three types of products that are produced from it: nutraceuticals, pharmaceuticals, and cosmetics [31]. Different parts of *G*. *lucidum* are commercially available, including mycelia, spores, and fruit body [6], and are sold as many different products, including powders, dietary supplements, and herbal tea [6,13]. Table 1 illustrates some of the commercial cosmetic products produced from *G*. *lucidum* mushrooms worldwide. Nowadays, the number of *Ganoderma*-based products well known commercially is estimated at over 100 brands [32]. The world trade market value of *G*. *lucidum* and its derivative products has reached approximately USD 4 billion [33]. 

#### 1.2.4. Scientometric Analysis

During the last decade, the *G*. *lucidum* mushroom has attracted multiple research fields, including biochemistry, genetics and molecular biology, agricultural and biological sciences, pharmacology, toxicology, pharmaceutics, and medicine. Figure 3 illustrates the increasing interest in multidisciplinary utilization of *G*. *lucidum* based on the number of research articles in the past 10 years.

### 1.3. Why Should Mushrooms, including Ganoderma lucidum, Be Considered Functional Foods?

#### 1.3.1. How to Define Functional Food?

In the early 1980s, the idiom “functional food” first appeared in Japan. Functional food is a broad term that includes several concepts [36]; for example, the definition of functional food provided by the Food and Agriculture Organization (FAO) states that “the functional food is the source that provides the human body with the necessary quantities of nutrients, i.e., proteins, carbohydrates, fats, vitamins, minerals, and others to keep it healthy. In addition, functional food can be cooked or prepared using ‘artificial intelligence technology’ [37]. In addition, the European Food Safety Authority (EFSA) defined functional food as “a food, which beneficially affects one or more target functions in the body, beyond adequate nutritional effects, in a way that is relevant to either an improved state of health and well-being and/or reduction of risk of disease” [38]. As described by the Functional Food Center (FFC) in the United States, functional foods are “real or processed foods that contain known or unknown biologically active compounds that, efficient, in defined and non-toxic quantities, recorded health benefit or provide a scientifically validated using unique biomarkers for the prevention, treatment or control of chronic disease or its symbiotic diseases” [39]. According to the definition of the Institute of Food Technologists (IFT), functional foods are those with ingredients that have health benefits in addition to basic nutrition, which is similar to the definition published by the International Life Sciences Institute (ILSI) [40,41]. 

Comprehensively, functional food can be defined as “a whole ingredient or a part of food that is used as food. It is part of a standard diet and is consumed on a regular basis, in normal quantities. It has proven health benefits that reduce the risk of specific chronic diseases or beneficially affect target functions beyond its basic nutritional functions” [42,43]. 

#### 1.3.2. What Do the Definitions of Functional Foods Conclude?

The use of the term “ingredient” means that functional food is not only conventional food but also could be a part of other food or food ingredients. In addition, the above-mentioned definitions of functional foods allow for adaptation to cultural differences, including widely differing “standards” among cultures and countries. Moreover, the use of the term “health benefits” is not restrictive. It refers to physiological, psychological, and biological advantages [43,44].

#### 1.3.3. Functional Foods and Their Relation with Gut Health

Among the important health effects of foods including the functional ones are those associated with gut health, a major determinant of an individual’s overall health. Several diseases are related in this context; e.g., gluten-therapy-resistant celiac, Crohn’s disease, ulcerative colitis, and irritable bowel syndrome. These adverse effects are caused by overgrowth and imbalance of intestinal bacteria linked to an individual’s food system [45]. The question that comes to mind is, what are the roles of the human gut in the body? These can be summarized as follows [45]:It converts food to nutrients;The human gut, via epithelial cell walls, assists in the absorption process of nutrients into the blood;The human gut inhibits toxic and strange particles from entering the bloodstream.

Consequently, and directly, any gut malfunction has adverse effects on human health. In this regard, functional foods, including pre- and probiotics, have become increasingly important due to their positive role in human gut health.

#### 1.3.4. *Ganoderma lucidum* as a Functional Food: How?

Historically, mushrooms, including *Ganoderma lucidum*, were traditionally consumed due to their nutritional and culinary values, and for their medical benefits when used in folk medicine. This historical heritage has recently been translated through molecular research to explore the present bioactive components and unlock mushrooms’ nutrition and therapeutic values [46,47]. Among these health benefits, mushrooms could help prevent diseases; e.g., hypertension, diabetes, hypercholesterolemia, and cancer, as mentioned in many reports. Hence, mushrooms can be considered a curative food [8,48]. Mushrooms are still untapped sources of bioactive substances such as glycoproteins, polysaccharides, mainly β-glucans, and secondary metabolites; i.e., nucleotide analogs, metal-chelating agents, terpenoids, polyphenols, alkaloids, lactones, and sterols. These biologically active components possess several therapeutic implications, such as antiviral, anticancer, hepatoprotective, immunopotentiating, and hypocholesterolemic agents [47,48,49,50,51]. 

The present paper critically discusses the benefits of *G*. *lucidum*, from nutritional value to medicinal impacts, and sheds light on its potential as a source of nutraceuticals and functional food. Moreover, this review provides answers with a critical vision to many questions, such as why the bioactive compounds of *G*. *lucidum* need to be further studied in vitro and in vivo, and what secrets are still behind them. Is it important to ensure *G*. *lucidum*’s quality and safety, as well as the best method to achieve that? With the potential of *G*. *lucidum*, will the future carry us to the possibility of commercial widescale use of *G*. *lucidum* and its products as new functional foods and medicines?

Despite that *Ganoderma lucidum* is not edible in its raw state due to its higher content of bitter compounds, its palatability can be increased by turning it into manufactured products such as powders, supplements, and tea [52]. The nutritional value of *G*. *lucidum* will be tackled in-depth in the following section. 

## 2. The Nutritional Profile of *Ganoderma lucidum*

“Medicines and food have a common origin”—Kaul [53].

For thousands of years, mushrooms have been valued throughout the world as food and medicine [8]. Nevertheless, mushrooms are still largely untapped resources in producing effective pharmaceutical products, nutrients, and cosmetics. Indeed, only approximately 150,224 species have been described [54] out of the estimated 2.2–3.8 million fungal species worldwide [55]. About 3000 species that belong to Macrofungi are safe for human consumption, such as edible mushrooms [56].

From the nutritionist’s point of view, generally, fresh mushrooms contain both soluble and insoluble fibers; the soluble fiber is mainly β-glucanpolysaccharides and chitosans [57]. However, a question comes to mind: does *G*. *lucidum* that is grown naturally or wild differ from that grown artificially in its nutritional components? According to the research in this regard, the answer is yes, as it was found that the quantities of crude protein, carbohydrates, and crude fiber were greater in the artificially grown variety [58]. Few studies have revealed the nutritional profile of *G*. *lucidum*. Roy and others [59] reported the nutritional value and mineral composition of *G*. *lucidum*. Through an analytical view of the nutritional profile of the *G*. *lucidum* mushroom (Table 2), several important conclusions can be reached.

*G*. *lucidum* contains a considerable amount of water-soluble proteins (19.5 g/100 g mushroom (*w*/*w*)). Moreover, 18 kinds of amino acids have been found in *G*. *lucidum*, and the most abundant amino acid was leucine, which possessed strong hypoglycemic and antioxidant activities [66,67].*G*. *lucidum* contains 3.5 g of dietary fiber per 100 g of mushroom (d/w).*G*. *lucidum* contains significant amounts of major minerals (e.g., phosphorus, sulfur) and other trace mineral contents; i.e., Cu, Mg, and Fe.As also mentioned in Table 2, *G*. *lucidum* is a highly rich source of vitamins such as riboflavin, niacin, thiamin, etc. Additionally, Ahmad [68] reported that several vitamins have been found in *G*. *lucidum*, such as vitamins B1, B2, B6, β-carotene, C, D, and E.Based on the nutritional profile of *G*. *lucidum*, this mushroom possesses a high nutrient potential that reflects positively on its health benefits.

Through this vision, the *G*. *lucidum* mushroom is increasingly becoming one of the natural and untapped medicine resources, which should be of interest to pharmaceutical, nutraceutical, and cosmetics manufacturers and consumers worldwide [69]. *Ganoderma lucidum* contains myriad biologically active compounds (over 400 compounds), including polysaccharides, triterpenoids, steroids, fatty acids, amino acids, nucleosides, proteins, and alkaloids [70]. Still, how do these bioactive compounds reflect their medical properties? The following section will discuss the therapeutic impacts of these bioactive compounds.

### 2.1. Ganoderma lucidum Is a Factory of Biologically Active Useful Compounds

“Mushroom of immortality & symbol of traditional Chinese medicine”—Chen et al. [71].

The biologically active molecules of *G*. *lucidum* rely on their chemical composition, with polysaccharides, peptidoglycans, and triterpenes being the three major bioactive compounds [58,68,70,71,72,73]. Additionally, this mushroom contains other constituents with distinct biological functions, such as minerals (e.g., germanium), proteins, lectins, crude fibers, phenols, enzymes, sterols, and long-chain fatty acids [6,74,75,76,77]. Table 3 shows the major bioactive compounds and their biological effects. ***Snapshots of these bioactive compounds could be found as follows.***

### 2.2. Polysaccharides and Peptidoglycans

Polysaccharides, such as ganoderans, represent diverse biological macromolecules with a broad range of biological properties [58]. Additionally, *G*. *lucidum* is a source of polysaccharides, glycopeptides, and polysaccharide crude extracts, as indicated by several studies [86]. In addition, these components of *G*. *lucidum* mushroom showed strong biological activities, including, for example, antioxidant, anti-tumor, and antibacterial activities due to its content of sugars, glycoproteins, and polysaccharide extracts obtained from the fruiting bodies [81,87,88,89]. Anti-inflammatory, hypoglycemic, antitumorigenic, and immunostimulating activities are among the multiple biological roles of polysaccharides extracted from *G*. *lucidum* [90,91,92,93,94,95]. Free radical scavenging abilities, reducing power, and chelating on ferrous ions are among the reported antioxidant properties [96,97]. Ospina et al. [98] reported that the isolated chitosan from *G*. *lucidum* has promising and desirable characteristics in specialized sectors such as biomedicine, pharmaceutics, and cosmetics, beyond the food industry. Regarding the peptidoglycans, *G*. *lucidum* contains a proteoglucan (GLPG) that has antiviral activity [99]. 

### 2.3. Triterpenes

Several triterpenes extracted from *G*. *lucidum* have been reported (around 100 types of triterpenes), with half of these types being novel and unique to *G*. *lucidum* [18]. Ganoderic and lucidenic acids are the major triterpenes produced by *G*. *lucidum*, while other triterpenes have been identified; e.g., ganodermic, ganoderiols, and ganoderal acids [58,100,101,102,103,104,105,106,107].

### 2.4. Other Bioactive Compounds

#### 2.4.1. Germanium

The element germanium has brought some attention to *G*. *lucidum*. Germanium is one of the most prevalent elements in wild *G*. *lucidum*. With 489 μg/g, germanium occupied the fifth-highest rank among the other detected minerals in terms of concentration [108]. This element possesses significant biological activities; i.e., antimutagenic, antitumor, immune-potentiating, and antioxidant [109]. There is no rigorous proof linking germanium with the specific health benefits of *G*. *lucidum*.

#### 2.4.2. Proteins

Some bioactive proteins purified from *G*. *lucidum* have been found to contribute to the medicinal properties of this mushroom; for example:LZ-8, an immunosuppressive protein [110];GLP, which possesses both antioxidant and hepatoprotective activities [111,112];Ganodermin, an antifungal protein [113].

Many other bioactive compounds have been isolated from *G*. *lucidum*, including:Enzymes; e.g., a metalloprotease that delays clotting time [6].

## 3. *Ganoderma lucidum* as a Functional Food 

For several hundred years, *G*. *lucidum* has been used to promote human health as a functional food through traditional treatment strategies. Nowadays, many published studies have established the multiple health benefits of *G*. *lucidum* in preventing or fighting multiple gastrointestinal and extraintestinal diseases, from constipation and gastritis, to anorexia, arthritis, asthma, bronchitis, and diabetes [35,75,95]. Additional studies have reported on the anticancer [6,31,52,114,115], preventing cardiovascular disease, and tumorigenesis [116,117,118,119], antioxidant [6,120,121], cardioprotective [122], antidiabetic potency [6,123,124], and antimicrobial activity [6,35,125] of this mushroom. Altogether, Figure 4 demonstrates the nutritional and health benefits of *G*. *lucidum*, ***which will be spotlighted individually as follows***.

### 3.1. Antimicrobial Activity

*G*. *lucidum* has been reported as a promising source of antimicrobial molecules (mainly polysaccharides) against various viral, bacterial, and fungal pathogens [79,83,125,126,127,128,129,130]. Table 4 summarizes the antimicrobial activities of the *G*. *lucidum* mushroom and its products.

### 3.2. Antiviral Potential

There have been few scientific studies (particularly on animals) that examined the antiviral effects of *G*. *lucidum* (Lingzhi); however, Zhu et al. [149] examined the anti-influenza effects of a hot water extract of Lingzhi on infected mice through intranasal and oral administration. The authors of this study concluded that short-term oral consumption of Lingzhi hot water extract had a limited effect in fighting influenza. Therefore, the authors recommended further study on the long-term anti-influenza effects that could improve the functional uses of this mushroom against influenza.

#### 3.2.1. *Ganoderma lucidum* against Enterovirus 71 (EV71)

Since 1969, “the same year in which the infection of human enterovirus 71 (EV71) infection was identified for the first time”, the infection mechanism has not been fully understood [150]. However, this viral infection was associated with several clinical diseases, ranging from neurological disorders to hand–foot–mouth disease (HFMD), and is considered a serious threat to children under six years old [151]. Currently, there are no certified prophylactic or therapeutic treatments for EV71 infection [152,153]. Outbreaks of EV71 infection have been periodically reported worldwide [154,155,156]. For instance, China has recently seen increased deaths linked to EV71 infection and HFMD among the young population [131,138,157,158]. As mentioned above, there are no approved drugs for preventing or treating EV71 infection, but currently, antiviral drugs with a broad spectrum (e.g., acyclovir, ganciclovir, and ribavirin) are used to partially relieve infection symptoms, although they have high cytotoxic side effects [159]. Therefore, investigation of novel and efficient medicines is urgently needed to control this severe viral infection. The adoption of natural medicinal compounds and Chinese herbal medicines has been observed across Asian countries for centuries, and recently in Western medicine [160,161]. *G*. *lucidum* is widely used as a folk medicine for a variety of ailments [162]. Zhang et al. [79] suggested that Lanosta-7,9(11),24-trien-3-one,15;26-dihydroxy (GLTA), and ganoderic acid Y (GLTB), which are triterpenoid compounds of *G*. *lucidum*, could prevent EV71 infection by interfering with the viral particle and limiting the viral adsorption to the host cells. Additionally, the interaction dynamics of GTLA and GLTA with the EV71 virion, predicted by molecular docking, showed potent molecular binding to the viral capsid protein at a hydrophobic pocket (F site), and hence a block uncoating of EV71 (Figure 5). Furthermore, it has been shown that GLTA and GLTB notably prohibited the viral RNA (vRNA) replication of EV71 by blocking EV71 uncoating. Therefore, both GLTA and GLTB may represent two promising curative agents to control and treat EV71 infection.

#### 3.2.2. *Ganoderma lucidum* against Dengue Virus (DENV)

The dengue virus (DENV), classified within the Flaviviridae family, is a fatal microbe transmitted to humans through mosquitoes (*Aedes albopictus* and *Aedes aegypti*) [163,164,165], causing both hemorrhagic fever [166,167], and shock syndrome [168,169]. A total of five different serotypes of DENV have been reported to induce both dengue fever types while potentially causing fatal infections [170,171]. Proteome analysis revealed that the translated DENV polyprotein complex comprises three structural and seven nonstructural proteins [171,172]. Of particular interest, the cofactor NS2B is required to fully activate the viral NS3 protease (NS3pro) domain that encodes a serine protease (S7 family). The NS2B–NS3pro complex of the dengue virus has been recently identified as an ideal target for developing novel anti-DENV drugs [173,174,175]. As one of the bioactive compounds extracted from *G*. *lucidum*, triterpenoids have been proposed and tested as antiviral agents against different viral pathogens; e.g., the human immunodeficiency virus. Ganodermanontriol, as a potent bioactive triterpenoid, was suggested to inhibit the DENV NS3pro protein based on in vitro studies. Thus, ganodermanontriol could act as a drug against DENV infection [176].

#### 3.2.3. *Ganoderma lucidum* against the 2019 Novel Coronavirus (SARS-CoV-2)

December 2019 marked in Wuhan (Hobby Province, China) the beginning of a mysterious pneumonia outbreak [177]. A month later, the infectious agent was revealed to be a new kind of coronavirus named SARS-CoV-2 (formerly 2019-nCOV) [178]. The World Health Organization (WHO) declared the pneumonia outbreak that appeared in Wuhan a major public health crisis on 11 February 2020 and gave it the official name of Coronavirus Disease-2019 (COVID-19) [179]. Multiple symptoms were reported in the COVID-19 patients, including cough, lung damage, fever, fatigue, muscle pain, diarrhea, myalgia, and respiratory symptoms [180,181]. As of 27 April 2021, 147,539,302 cases of SARS-CoV-2 infected pneumonia and 3,116,444 deaths had been reported in China and 223 other countries, areas, or territories, of which 103,503 cases were found in China [182]. Natural products are among the most important sources for modern medication industry technology, if not the most important, due to their advantages such as abundant clinical use, and their unique diversity of chemical structures and biological activities [183,184]. In this context, traditional Chinese medicine (TCM) is one of the gold mines rich in untapped natural resources [185,186] that can be employed to treat many diseases that represent a challenge for humankind, including COVID-19. The previous studies on SARS-CoV and its homology with SARS-CoV-2 may provide avenues to natural compounds that inhibit SARS-CoV-2 [187]. For instance, the helicase domain is being investigated as a possible drug target. Yu et al. [188] reported that scutellarein and myricetin potently prevented nsP13, a SARS-CoV helicase protein, in vitro by altering its ATPase activity. The RNA-dependent RNA polymerase is another potential target for developing antiviral compounds, being an essential enzyme for RNA synthesis. Indeed, dose-dependent inhibition of this SARS-CoV enzyme was reported for the extracts of *G*. *lucidum* (IC50:41.9 µg/mL), *Coriolus versicolor* (IC50:108.4 µg/mL), *Sinomenium acutum* (IC50:198.6 µg/mL), and Kang Du Bu Fei Tang (IC50:471.3 µg/mL) [189]. Therefore, *G*. *lucidum* could serve as a novel and promising source of bioactive natural compounds with anticoronavirus activity [187].

### 3.3. Antioxidant and Antiaging Activity

Multiple research studies reported a close relationship between the richness of *G*. *lucidum* in “phenolic compounds, triterpenes, polysaccharides, polysaccharide peptide” and its antioxidant biological activity [83,97,190,191,192]. Clinical nutritionists have demonstrated that consuming antioxidant-rich plant-based foods may protect from cancer and many other chronic diseases [193,194]; however, this causality is still not proven yet for the antioxidants of *G*. *lucidum* [195]. Hence, one of the research priorities for the *G*. *lucidum* mushroom is to conduct more studies to close the gap in the interplay between antioxidants and the host immune system [191].

The long-term presence of free radicals and reactive oxygen species (ROS) accelerates aging and numerous age-associated illnesses [13]. Therefore, studies on scavenging free radicals and ROS are particularly important in antiaging research. *G*. *lucidum* polysaccharides (GLPs) can inhibit ROS production in fibroblasts following UVB treatment [196].

### 3.4. Anticancer Activity

Cancer is still one of the most fatal diseases worldwide and poses a major clinical challenge despite the notable boom in early diagnostic techniques and evolution in its treatment techniques [197]. Hundreds of plant species have been investigated as sources for new therapeutics (chemopreventive or chemotherapeutic) [198]. In this regard, mushrooms; e.g., *Ganoderma* species, are rich sources of many biologically active components, including antitumoral agents [199,200]. For example, polysaccharides and triterpenes are two major groups of compounds extracted from *G*. *lucidum* that were reported to possess chemopreventive and/or tumoricidal activities [6,31,52,114,115,201,202,203]. In addition, the antitumor activity exhibited by *G*. *lucidum* is achieved via induction of programmed cell death, as reported by many studies [81,204,205]. Moreover, the isolated compounds from *G*. *lucidum* have been previously described as modulators of autophagy in numerous human tumor cell lines [206,207,208,209]. In the same context, a methanolic extract (extraction at room temperature) of *G*. *lucidum* fruiting bodies prevented the growth of a human gastric tumor cell line via a mechanism that involved cellular autophagy [209]. Still, it is unknown if the extract is an inducer of autophagy or an autophagic flux inhibitor. More recently, Reis et al. [210] demonstrated that a methanolic extract of *G*. *lucidum* caused autophagy induction, rather than reducing the autophagic flux in AGS cells.

### 3.5. Antidiabetic Activity

*G*. *lucidum* has been proved to possess compounds responsible for hypoglycemic effects, such as polysaccharides, proteoglycans, proteins, and triterpenoids [6,78,123,124]. For instance, Wang et al. [211] reported that consuming a *G*. *lucidum* spore powder (GLSP) induced a decrease in blood glucose levels by promoting glycogen synthesis and preventing gluconeogenesis.

### 3.6. Cardioprotective Effects

How does *G*. *lucidum* have cardioprotective impacts? Many studies have answered this question. Firstly, Sudheesh et al. [122] reported the presence of α-tocopherol in *G*. *lucidum* that protected the mitochondria, reducing cardiac toxicity and mitochondrial dysfunction. Additionally, Gao et al. [212] referred to the positive effects of ganopoly (*G*. *lucidum* polysaccharide extract) on coronary heart disease (CHD) patients. The same authors showed that a polysaccharide extract of *G*. *lucidum* induced decreased blood pressure and serum cholesterol levels.

### 3.7. Hepatoprotection

The GLPs and *Ganoderma* triterpenoids (GTs) can act on the immune system and effectively exhibit hepatoprotective effects and treat liver damage. The hepatoprotective effects of *G*. *lucidum* have been widely studied [213]. GLPs can protect hepatocyte injury by inhibiting lipid peroxidation, elevating antioxidant enzyme activity, and suppressing apoptosis and immune-inflammatory response [214]. GTs offered significant cytoprotection against the oxidative damage induced by tertbutyl hydrogen peroxide (t-BHP) in hepatocellular carcinoma cells by decreasing the level of malondialdehyde and increasing the contents of glutathione and superoxide dismutase (SOD) [215]. Analysis of histopathology and serum enzymes in mice revealed an important hepatoprotective function of an ethanol extract of *G*. *lucidum* (GLE). It was therefore assumed that GLE could improve alcohol-induced liver injury [216]. In addition, a *G*. *lucidum* mycelium-fermented liquid (GLFL) was reported to possess hepatoprotective properties in rats [217].

### 3.8. Anti-Inflammatory Effects

Inflammation is a normal physiological response to an infection or injury and is part of host defense and tissue healing [218]. GLPs can prevent inflammation, maintain intestinal homeostasis, and regulate the intestinal immunological barrier functions in mice [219]. The anti-inflammatory effect of GLPs plays an important role in the care of sensitive skin [213].

### 3.9. Prebiotic Potential

Prebiotics are defined as “a substrate that is selectively utilized by host microorganisms conferring a health benefit” [220,221]. Mushrooms are considered untapped sources of prebiotics such as fibers, oligosaccharides (major constituents of mushrooms), and polyphenols, which can boost the growth and metabolic activity of beneficial members of the gut microbiota. For example, nondigestible polysaccharides can prevent pathogen proliferation by improving the growth of probiotics in the gut [222]. During the last decade, the interplay between prebiotics and human gut microbiota and its implications in mitigating many diseases; e.g., cancer, diabetes, and obesity gained much focus and has emerged as one of the principal trending axes of food science and technology. Scientific evidence has accumulated on the critical role of gut microbiota dysbiosis in exacerbating inflammation in host tissues, from the intestinal environment to the brain. Likewise, critical data has established the gut microbiota’s regulatory role in energy metabolism, which may cause disturbances in the metabolism processes [223]. For instance, mushrooms are a rich source of prebiotics that may play a pivotal role in treating pneumonia and atherosclerosis, as well as in their antitumor activity [224]. In the same context, a study conducted on mice (C57BL/6) confirmed that the Mexican *G*. *lucidum* is a rich source of prebiotics that reduced blood cholesterol [225]. The same study attributed the ability of Mexican *G*. *lucidum* to lower the blood cholesterol level to the significant decrease in the lipid-generating gene expression (*Hmgcr*, *Fasn*, *Srebp1c*, *Acaca*), and *Abcg5*, *Abcg8* as genes responsible for reverse cholesterol transport, simultaneous with an increase in *Ldlr* gene expression in the liver [225]. Another study showed the possibility that *G*. *lucidum* polysaccharide peptides (GLPP) may have a role in alleviating the disturbance in the metabolism of fats through the ability of these compounds to alter the composition of the gut microbiota, which in turn has a positive effect on controlling and reducing the disruption of fat metabolism, regulating genes involved in intestinal integrity, bile acid homeostasis, and extrauterine fat deposition (Figure 6). Thus, GLPP can be considered a potential functional food component for treating hyperlipidemia and gut microbiota dysbiosis [226].

Furthermore, to date, no extensive studies have examined the biological activities and functions of GLFL on the regulation of the gut microbiota and cardiovascular diseases (CVDs) [227].

Concerning this, it has been demonstrated that the gut microbiota could play an important role in host health through their influence on cardiovascular risk factors [228,229]. As was mentioned previously, the products associated with *G*. *lucidum* have positive effects on the gut microbiota, and thus these products can regulate the risk factors for cardiovascular disease in the intestine. Chang et al. [230] reported an altered gut microbiota composition in an obese mouse model treated with a water extract of *G*. *lucidum* mycelium. In addition, GLFL was shown to reduce plasma low-density lipoprotein (LDL-c) cholesterol, triglycerides, and total cholesterol, and increase high-density lipoprotein (HDL-c) cholesterol in mice [231]. Additionally, Wu et al. [227] reported that when GLFL was fed to humans, it profoundly altered the gut microbiota. In the post-feeding group, there was an evident difference in β diversity as compared to the case of the pre-feeding group; this suggested that GLFL had altered the composition of the gut microbiota. Furthermore, the same authors reported that GLFL could protect humans by stimulating the growth of probiotics (i.e., genus *Lactobacillus* (*p* < 0.05)) while inhibiting the growth of pathogens (i.e., genus *Aggregatibacter* and *Campylobacter* (*p* < 0.05)).

### 3.10. The Health Risks of Ganoderma lucidum and Its Products

Most research on *G*. *lucidum* and its products has reported positive clinical outcomes and potential therapeutic uses, while the innocuity and potential toxic effects in humans have been poorly investigated. For instance, *G*. *lucidum* extracts could cause toxicity in vitro [35]. Moreover, *G*. *lucidum* spore-powder treatment caused hepatotoxic effects, as reported by Wanmuang et al. [232] Although no adverse effects of consumption of *G*. *lucidum* on lactation were proven, it is not advised for pregnant or lactating women [233].

## 4. Future Trends

Thus far, many questions have arisen regarding why and how to expand and maximize the utilization of *G*. *lucidum* and its derivatives, and what to do about the new applications and innovative techniques used in this regard. ***These points will be discussed* via *three axes as follows.***

### 4.1. Do the Beneficial Medical Properties of G. lucidum Need More Scientific Evidence?

Several publications have reported that *G*. *lucidum* may have diverse beneficial therapeutical characteristics via its myriad bioactive compounds, such as triterpenes, polysaccharides, and proteins; hence, *G*. *lucidum* and its products are still common as commercial products. Therefore, the efficacy and safety of the consumption of *G*. *lucidum* are still considered knowledge gaps that are poorly investigated. During the past three decades, in vitro/in vivo studies reported by Western researchers have shown biomedical benefits for *G*. *lucidum*, which helped promote this mushroom in the Western world. However, there still is an urgent need to fully understand the related biomechanisms, and thus unlock their biotherapeutic application. The isolation, purification, and identification of active compounds of *G*. *lucidum* should be carried out to decipher the bioactivity of these compounds within its nutraceutical and pharmaceutical products. This aspect is a big challenge when implementing the commercial standardization strategies of *G*. *lucidum* products [6,35].

More research is needed to re-examine and study the bioactive ingredients extracted from *G*. *lucidum*, and this will be beneficial for clinical applications due to the discrepancy in the research results for *G*. *lucidum* or its derivative products (e.g., GLFL, GLPP, and WEGL). For example, Wu et al. [231] reported that a GLFL was unsafe because it increased the number of opportunistic pathogens; e.g., *Acinetobacter*, and decreased probiotics; e.g., *Lactococcus*. These results were in contrast to what was obtained when they were applied to mice [226].

Continued genetic studies of *G*. *lucidum* will elucidate the biosynthesis of the therapeutically active compounds produced by this mushroom; e.g., the unique triterpenoid antitumor ganoderic acids (GAs). The clustered regularly interspaced short palindromic repeats and CRISPR-associated protein 9 (CRISPR-CAS9) technology has positively identified active curative components in *G*. *lucidum* by constructing functional genes of GA biosynthesis in this mushroom, thus serving as a vital platform for metabolic engineering in *G. lucidum*. Therefore, the CRISPR-CAS9 technique can be a cornerstone in all biotechnological applications of *G*. *lucidum*, such as molecular breeding. Therefore, a complete understanding of the *G*. *lucidum* genome will pave the way for its future roles in medical and industrial applications [71,115,234].

Large-scale studies on *G*. *lucidum* mushrooms will be conducted with standard scientific methods in the near future.

### 4.2. Future of the G. lucidum Mushroom in the Food Industry

Nowadays, several *Ganoderma lucidum*-based products are available in nutraceutical form. Some of them are marketed as dietary supplements and are widely consumed in many countries such as the United States, where they are combined with many other ingredients; e.g., coffee and tea. Because there is no proper toolkit, the consistency of the quality of dietary supplements derived from *G*. *lucidum* is rarely evaluated. Additionally, *G*. *lucidum* could be considered a source of food preservatives [5,187,235].

To validate *G*. *lucidum*’s nutraceutical usage, more research on this mushroom is needed.

### 4.3. Is Tracing the Species and Geo-Origin of G. lucidum Essential?

Research conducted by Loyd and others [14] showed that manufactured *G*. *lucidum*-based products (e.g., dietary supplements), which are marketed as derived from *G*. *lucidum,* contain not only *G*. *lucidum* but also multiple *Ganoderma* species, are unfortunately sold for medicinal uses. Of course, not all *Ganoderma* species produce the same therapeutic compounds, the same quality, or the same quantities. This raises questions about the traceability and authenticity of mushroom species, and how important this is in the industry. Therefore, this question should be addressed in subsequent research focusing on *G*. *lucidum* and its products.

As mentioned by Qi et al. [236], the geographical-origin traceability of mushrooms and their products is critical to assure their quality and safety. Indeed, the nutritional and therapeutic properties of each mushroom species vary depending on the geo-origins [237,238,239,240]. Lu et al. [237] proved this fact through their research on samples of *G*. *lucidum* collected from different geographic regions, in which they found that the content of each *G*. *lucidum* sample of ganoderic acids A and B, polysaccharides, and triterpenoids varied according to their geographical origin (including the differences in cultivation and environmental conditions). Hence, the geo-origin traceability of *G*. *lucidum* will reinforce the value of this mushroom globally at all levels, whether industrial or economic.

Then, what is the best method that can be used for species and geo-traceability targets? El Sheikha and Hu [8] proposed the DNA barcoding approach as a new “cutting edge” technology to significantly enhance food traceability in general and in mushrooms, especially from the field to the table.

## 5. Infographic for *Ganoderma lucidum*: Current Scenario and Future Perspectives

Recently, research on *G*. *lucidum* and its products has achieved substantial progress and has become a focus of the attention of the scientific community in many fields. Many studies from different viewpoints elucidated the biological characteristics, chemical composition and active components, pharmacological effects and related mechanisms, and clinical applications based on *G*. *lucidum*. Furthermore, at the industrial level, *G*. *lucidum* has made some progress.

In the future, new chemical compositions and active components (as a promising functional food), cellular and molecular mechanisms of biological activities (e.g., prebiotic effects), rapid and confirmatory methods to identify effective ingredients, fermentation and cultivation techniques, double-blind large-scale clinical trials, and quality control monitoring of product will be the aims of *G*. *lucidum* research (see Figure 7).

## 6. Conclusions

*Ganoderma lucidum* (Lingzhi, Reishi, or Mannentake) is a promising source of prebiotics due to its abundance of several bioactive compounds that have nutritional and medicinal effects and are present in all parts of the fungus (fruit bodies, mycelium, and spores). Therefore, since ancient times, *G*. *lucidum* has been used traditionally in Chinese medicine to treat chronic diseases. In addition, Chinese tradition refers to *G*. *lucidum* as “the lucky fungus” for its power to alleviate conditions such as arthritis, insomnia, and chest tightness.

There has been an increased interest in *G*. *lucidum* as a dietary supplement containing Reishi, which is a widespread therapeutic agent worldwide. As for Western countries, the bioactive substances extracted from *G*. *lucidum* have been used in alternative medicine to support traditional medicine in treating severe diseases, including diabetes, hepatitis, and cancer. Nevertheless, continuing this trend requires more clinical trials, typically to confirm efficacy and safety. Soon, more studies will be conducted on this mushroom on a larger scale in terms of medicinal applications or the food industry. The geographical origin is considered one of the critical factors that greatly impact both the safety and quality of mushrooms. Therefore, the determination of geographical origin has become an essential requirement to provide consumers with safe and high-quality mushrooms, including *G*. *lucidum*. Although there are many challenges facing the production of nutraceuticals and functional foods from *G*. *lucidum* on a large scale, especially in light of the limited clinical trials in humans, there is a potential for innovation, development, and expansion of applications (e.g., in food and pharmaceutical applications) due to *G*. *lucidum*’s promising nutritional and health characteristics.

## Figures and Tables

**Figure 1 foods-11-01030-f001:**
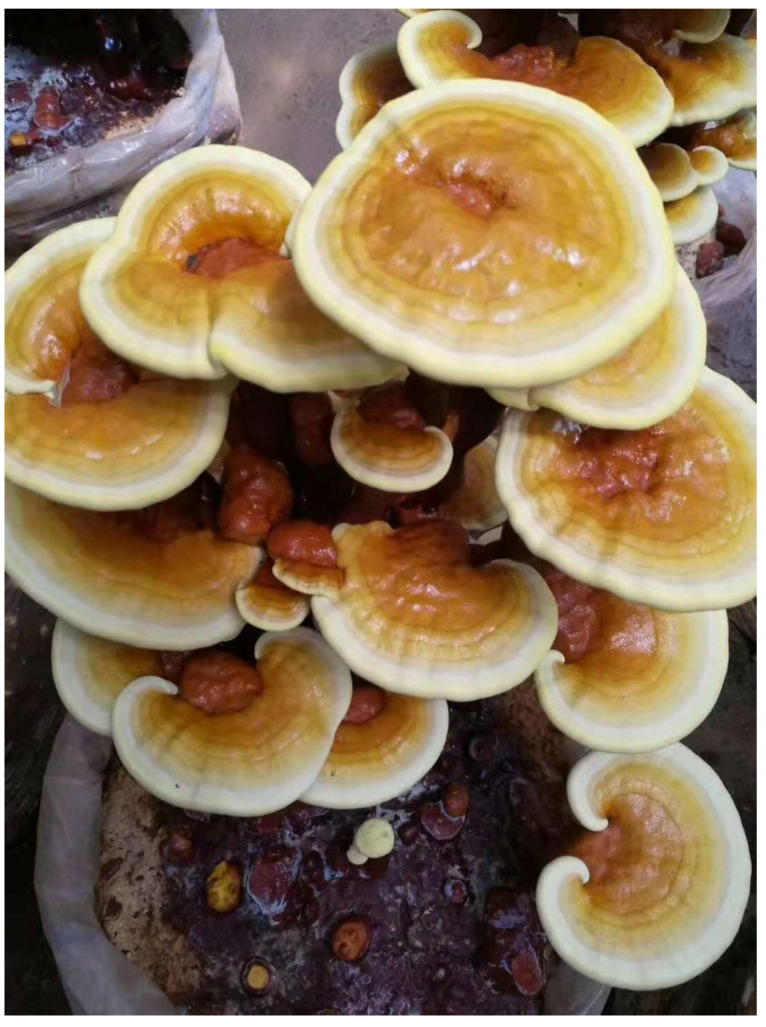
Image of *Ganoderma lucidum*.

**Figure 2 foods-11-01030-f002:**
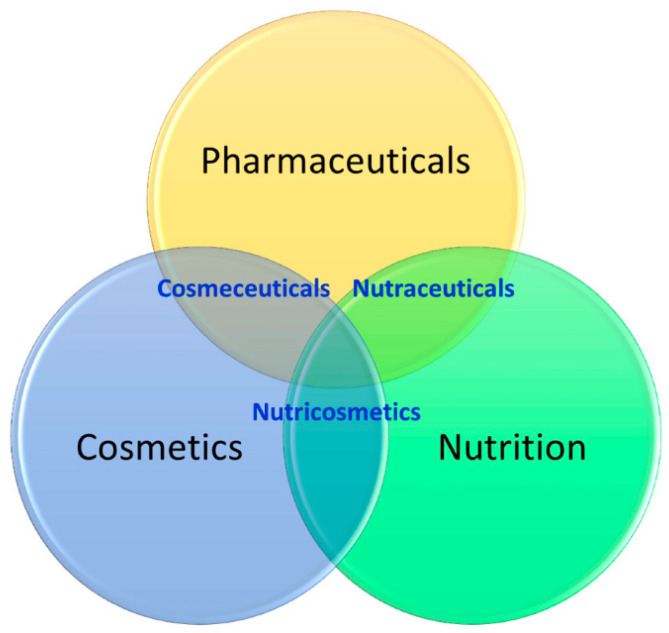
Wide-scale applications of mushrooms including *Ganoderma lucidum*; i.e., pharmaceuticals, nutraceuticals, and cosmetics. Source: Reprinted from Wu et al. [31]. Licensed under CC BY 4.0.

**Figure 3 foods-11-01030-f003:**
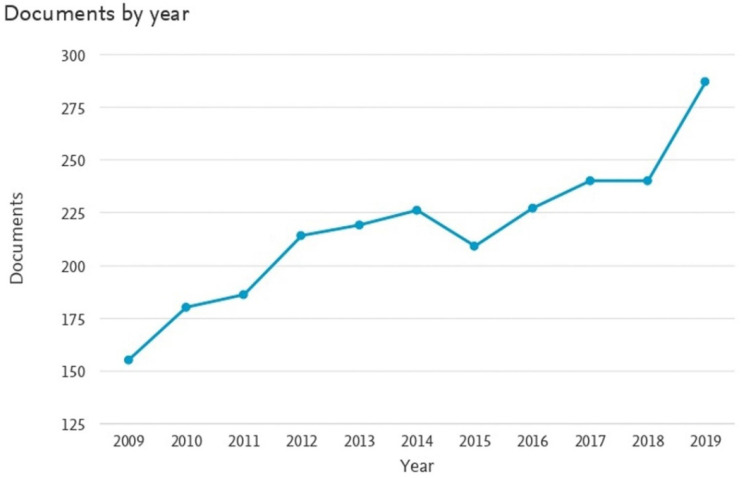
A scientometric analysis of increasing interest in *Ganoderma lucidum* over the last 10 years. Reprinted with permission from Scopus. 2020, Elsevier.

**Figure 4 foods-11-01030-f004:**
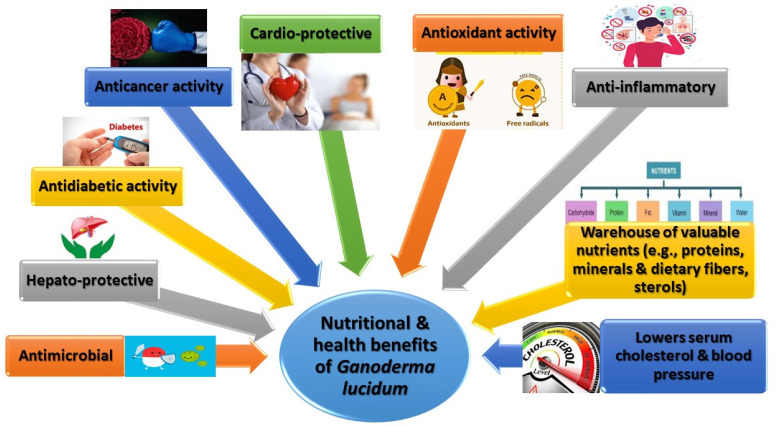
Nutritional and health benefits conferred by *Ganoderma lucidum*.

**Figure 5 foods-11-01030-f005:**
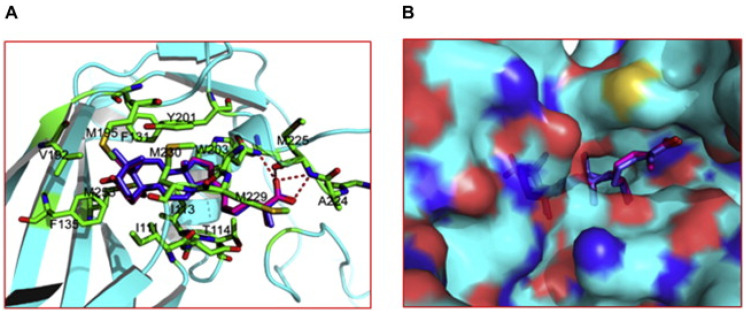
Molecular docking for the interaction of antiviral compounds with EV71 capsid. (**A**) Stick conformer diagram. (**B**) Cartoon conformer diagram. Both GLTA and GLTB could bind stably in the viral capsid mainly through hydrophobic interactions at a hydrophobic pocket (F site) in the capsid of EV71 virion. Source: Reprinted with permission from Zhang et al. [79]. 2022, Elsevier.

**Figure 6 foods-11-01030-f006:**
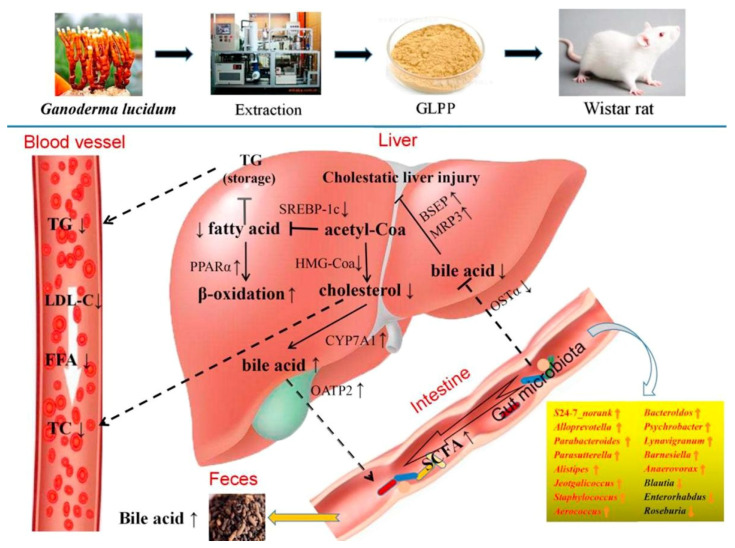
Regulatory mechanism of GLPP on hyperlipidaemia, hypercholesterolemia, and gut microbiota dysbiosis in rats fed on HFD. GLPP: *Ganoderma lucidum* polysaccharide peptide; HFD: high-fat diet; TG: triglyceride; TC: total cholesterol; LDL-C: low-density lipoprotein cholesterol; FFA: free fatty acids; SCFA: short-chain fatty acids; OSTα: organic solute transporter alpha; CYP7A1: cholesterol 7α-hydroxylase; SREBP-1C: sterol regulatory element-binding protein-1C; PPARα: peroxisome proliferator-activated receptor alpha; HMG-Coa: 3-hydroxy-3-methylglutaryl coenzyme A; BSEP: bile salt export pump; MRP3: multidrug-resistance-associated protein 3; OATP2: organic-anion-transporting polypeptides 2. Source: Reprinted with permission from Lv et al. [226]. 2022, Elsevier.

**Figure 7 foods-11-01030-f007:**
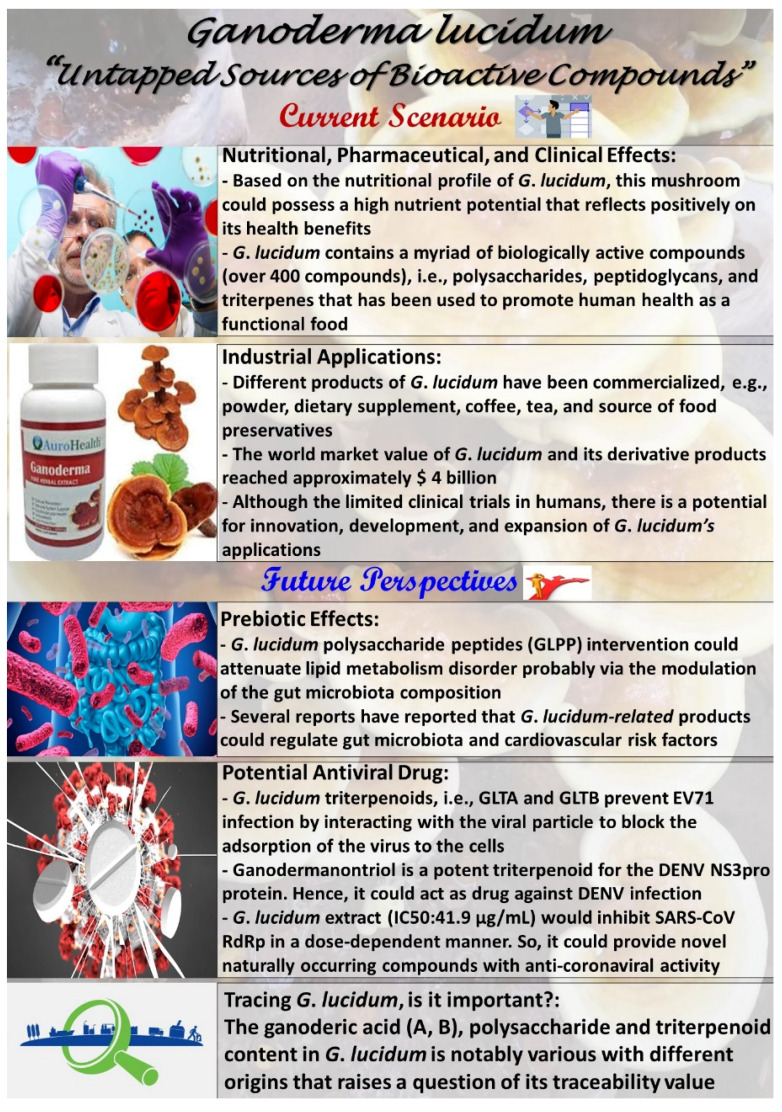
Infographic for *Ganoderma lucidum*: current scenario and future perspectives.

**Table 1 foods-11-01030-t001:** Some of the cosmetic products are produced commercially from the *G*. *lucidum* mushroom worldwide *.

Commercial Product Name/Producing Country	Uses
CV Skinlabs Body Repair Lotion, USA	Wound healing and anti-inflammatory
Dr. Andrew Weil for Origins Mega-Mushroom Skin Relief Face Mask, USA	Anti-inflammatory properties
Moon Juice Spirit Dust, USA	Immune system
Estée Lauder Re-Nutriv Sun Supreme Rescue Serum sun care product, USA	Triple-action repair technology to enhance the skin’s own natural defenses against the visible effects of sun exposure and sun-stressed skin
Four Sigma Foods Instant Reishi Herbal Mushroom Tea, UK	Immunity boost
Kat Burki Form Control Marine Collagen Gel, UK	Boosting collagen, improving elasticity, and providing hydration
Tela Beauty Organics Encore Styling Cream, UK	Providing hair with sun protection and preventing color fading
Menard Embellir Refresh Massage, France	Skin antiaging
Yves Saint Laurent Temps Majeur Elixir De Nuit, France	Antiaging
Pureology NanoWorks Shineluxe, France	Antiaging and antifading
Hankook Sansim Firming Cream (Tan Ryuk SANG), Korea	Making skin tight and vitalized
La Bella Figura Gentle Enzyme Cleanser, Italy	Antioxidants and vitamin D
DXNGanozhi Moisturizing Micro Emulsion, Malaysia	Hydrating and nourishing the skin
Guangzhou Bocaly Bio-Tec. *Ganoderma* Cell-Repairing Antiaging Face Mask, China	Antiwrinkle, firming, lightening, moisturizer, and nourishing, pigmentation corrector; pore cleaning and whitening
Nanjing Zhongke Pharmaceuticals Ganoderma Face Cream Set (day/night cream and eye gel set), China	Immunity boost and antifatigue
Shenzhen Hai Li Xuan Technology HailiCare Skin Whitening Cream, China	Removing freckles and whitening
Menard Embellir Night Cream, Japan	Eliminating toxins and helping repair skin damage associated with overexposure to UV radiation and free radicals
MAVEX Rejuvenating Treatment, Hong Kong	Antioxidant action and deep cellular renewal; fight degenerative processes and the negative action of free radicals

* Sources: Wu et al. [31], Taofiq et al. [34], Hapuarachchi et al. [35], www.vegamebeljepara.com, www.dazzlinggroup.com, www.dxnmalaysia.com, and www.vegamebeljepara.com (accessed on 16 February 2022) Adapted from Wu et al. [31]. Licensed under CC BY 4.0. Adapted with permission from Taofiq et al. [34]. 2022, Elsevier.

**Table 2 foods-11-01030-t002:** Physicochemical properties and chemical composition of *Ganoderma lucidum* mushroom.

Constitute	Content			DRIs * (g/day)	Value in 100 g Mushroom/DRIs × 100
	Value	g/100 g Mushroom (Wet-Weight Basis)	g/100 g Mushroom (Dry-Weight Basis)		
Moisture %		47			
Total solids (TS) %			53		
pH value	5.6				
Energy (kcal)	238.98 **			Men: 2215 ***	10.79
				Women: 2025	11.80
Water-soluble proteins %		19.5	36.80	Men (total proteins) ****: 56	34.82
				Women (total proteins): 46	42.39
Total lipids %		3.00	5.66	44–77 *****	3.90–6.82
Total ash %			6.3		
Reducing sugars %		4.39	8.28		
Nonreducing sugars %		1.02	1.92		
Total sugars %		5.41	10.21	130	4.16
Crude fibers %			3.5	Men: 38	9.21
				Women: 25	14.00
Polyphenols “as gallic acid”		0.04	0.08	1 ******	7.5
Mineral	Mineral content (mg/100 g mushroom)			DRIs (mg/day)	Value in 100 g mushroom/DRIs × 100
*Major minerals*					
Potassium	432			4700	9.19
Phosphorus	225			700	32.14
Sulfur	129			200–1500	8.60–64.50
Magnesium	7.95			Men: 400	2.00
				Women: 310	2.60
Sodium	2.82			1500	0.20
Calcium	1.88			1000	0.20
*Trace minerals*					
Copper	26			0.9	2889
Manganese	22			Men: 2.3	956.52
				Women: 1.8	1222.22
Iron	2.22			Men: 8	27.75
				Women: 18	12.33
Zinc	0.7			Men: 11	6.40
				Women: 8	8.75
Vitamin	Vitamin content (mg/100 g mushroom)			DRIs (mg/day)	Value in 100 g mushroom/DRIs × 100
Thiamine (B1)	3.49			Men: 1.2	290.83
				Women: 1.1	317.27
Riboflavin (B2)	17.10			Men: 1.3	1315.38
				Women: 1.1	1554.54
Niacin (B3)	61.9			Men: 16	386.87
				Women: 14	442.14
Pyridoxine (B6)	0.71			Men: 1.4	50.71
				Women: 1.2	59.16
Ascorbic acid	32.2			Men: 90	35.77
				Women: 75	42.93

* DRIs: dietary recommended intakes for adults [60,61]; ** the total energy of 100 g of mushroom samples was calculated according to the equations of Manzi et al. [62]; *** based on 1.3 kcal/kg body weight/hour for the reference body weight; **** based on 0.8 g/kg body weight/day for the reference body weight; ***** Casselbury [63]; ****** Duthie et al. [64]. Sources: Roy et al. [59], Rahman et al. [65], and http://www.medicinabiomolecular.com.br/biblioteca/pdfs/Biomolecular/mb-0223.pdf. (accessed on 16 February 2022). Adapted from Rahman et al. [65]. Licensed under CC-BY.

**Table 3 foods-11-01030-t003:** The major bioactive compounds of *G*. *lucidum* and their biological effects.

Bioactive Compounds	Biological Effects	References
*Triterpenoids*		
Ganoderic acids, lucidumol, lucialdehyde, lucidenic acids, ganodermic, ganolucidic acids, ganoderals, ganoderiols	Anticancer	Wachtel-Galor et al. [6], El Mansy [75]
Triterpenoids	Antidiabetic	Ahmad [68], Ma et al. [78]
Ganoderic acids T-Q and lucideinic acids A, D2, E2, and P	Anti-inflammatory	El Mansy [75]
Triterpenes	Antioxidant	El Mansy [75]
Ganoderic acids, ganodermin, ganoderic acid A, ganodermadiol, ganodermanondiol, lucidumol B, ganodermanontriol, ganoderic acid B, ganolucidic acid B	Antimicrobial	Cör et al. [70], Sudheer et al. [73]
Triterpenoids, ganoderic acid, ganoderiol F, ganodermanontriol	Antiviral	Bishop et al. [13], Zhang et al. [79], Zhu et al. [80]
*Polysaccharides*		
1→3, 1→4, and 1→6-linked β and α-D (or L)-glucans, GLP-2B	Anticancer	Wachtel-Galor et al. [6], Ferreira et al. [81]
Polysaccharides	Antidiabetic	Ahmad [68], Ma et al. [78]
Polysaccharides	Antioxidant	El Mansy [75]
Polysaccharides	Antimicrobial	Cör et al. [70]
Polysaccharides (ganopoly)	Cardiovascular problems	Chan et al. [82]
*Proteins, Glycoproteins, and Peptidoglycans*		
Glycopeptides and peptidoglycans	Anticancer	Wachtel-Galor et al. [6], Sudheer et al. [73], Ferreira et al. [81],
Protein Ling Zhi-8 (LZ-8), lectin, ribosome-inactivating proteins, antimicrobial proteins, glycopeptides/glycoproteins, peptidoglycans/proteoglycans, ganodermin A, ribonucleases, proteinases, metalloproteases, laccases	Immunomodulatory, anticancer, and antitumor	Wachtel-Galor et al. [6], El Mansy [75]
Proteoglycans, proteins (LZ-8)	Antidiabetic	Ahmad [68], Ma et al. [78]
Polysaccharide–peptide complex	Antioxidant	Mehta [83]
*Phenolic compounds*		
Phenolic components, phenolic extracts	Antioxidant	Mehta [83]
Saponins	Anticancer and antioxidant	Lee et al. [84]
Sterols; e.g., ergosterol	Provitamin D2	Wachtel-Galor et al. [6]
Long-chain fatty acids	Antitumor	Gao et al. [85]

**Table 4 foods-11-01030-t004:** Antimicrobial activities of *Ganoderma lucidum* parts, products, and compounds.

Parts/Products/Compounds	Tested Microorganism	References
Antibacterial activity		
Fruiting bodies	*Helicobacter pylori* ATCC 43504, *Staphylococcus aureus* ATCC 26003	Liu et al. [131], Shang et al. [132]
Mycelia extract	*Bacillus cereus* (clinical isolate), *Micrococcus flavus* ATCC 10240, *S*. *aureus* ATCC 6538, *Listeria monocytogenes* NCTC 7973, *Escherichia coli* ATCC 35218, *Enterobacter cloacae* (human isolate), *Pseudomonas aeruginosa* ATCC 27853, *Salmonella typhimurium* ATCC 13311	Ćilerdžić et al. [133]
Fruiting bodies	*S*. *aureus* (MTCC 96), *B*. *cereus* (MTCC 430), *P*. *aeruginosa* (MTCC 424)	Karwa and Rai [134]
Fruiting bodies	*S*. *aureus* (ATCC 6538), *Bacillus subtilis* (ATCC 6633)	Ćilerdžić et al. [135]
Ergosta-5,7,22-trien-3β-yl acetate, ergosta-7,22-dien-3β-yl acetate, ergosta-7,22-dien-3-one, ergosta-7,22-dien-3β-ol, ergosta-5,7,22-trien-3β-ol, ganodermadiol	*S*. *aureus* (ATCC 6538), *B*. *subtilis* (ATCC 6633)	Ćilerdžić et al. [135]
Carpophores	*Bacillus anthracis* ATCC 6603, *B*. *cereus* ATCC 27348, *B*. *subtilis* ATCC 6633, *Micrococcus luteus* ATCC 9341, *S*. *aureus* ATCC 25923, *E*. *coil* ATCC 259 22, *Klebsiella oxytoca* ATCC 8724, *Klebsiella pneumonia* ATCC 10031, *Proteus vulgaris* ATCC 27853, *S*. *typhi* ATCC 6229	Yoon et al. [136]
Basidiocarps	*B*. *cereus* (clinical isolate), *M*. *flavus* ATCC 10240, *S*. *aureus* ATCC 6538, *L*. *monocytogenes* NCTC 7973, *E*. *coli* ATCC 35218, *E*. *cloacae* (human isolate), *P*. *aeruginosa* ATCC 27853, *S*. *typhimurium* ATCC 13311	Vazirian et al. [137]
12b-acetoxy-3β,7 β -dihydroxy- 11,15,23-trioxolanost-8-en-26-oic acid butyl ester	*S*. *aureus* (ATCC 6538), *B*. *subtilis* (ATCC 6633)	Yang et al. [138]
Mycelia (Protein extract)	*Staphylococcus epidermidis*, *B*. *subtilis*, *B*. *cereus*, *E*. *coli*, *P*. *aeruginosa*	Sa-Ard et al. [139]
Fruiting bodies (Protein extract)	*S*. *epidermidis*, *S*. *aureus*, *B*. *subtilis*, *B*. *cereus*, *E*. *coli*, *P*. *aeruginosa*	Sa-Ard et al. [139]
NG *	*S. aureus* (ATCC 6538)*, B. cereus* (clinical isolate), *L. monocytogenes* (NCTC 7973), *M. flavus* (ATCC 10240), *P. aeruginosa* (ATCC 27853), *E. coli* (ATCC 35210), *S.* *typhimurium* (ATCC 13311), *E. cloacae* (human isolate)	Heleno et al. [140]
Antifungal activity		
Fruiting bodies	*Acremonium strictum* BEOFB10m, *Aspergillus glaucus* BEOFB21m, *Aspergillus flavus* BEOFB22m, *Aspergillus fumigatus* BEOFB23m, *Aspergillus nidulans* BEOFB24m, *Aspergillus niger* BEOFB25m, *Aspergillus terreus* BEOFB26m, *Trichoderma viride* BEOFB61m	Vazirian et al. [137]
Fruiting bodies	*A*. *fumigatus* (human isolate), *Aspergillus versicolor* (ATCC 11730), *Aspergillus ochraceus* (ATCC 12066), *A*. *niger* (ATCC 6275), *T*. *viride* (IAMz5061), *Penicillium funiculosum* (ATCC 36839), *Penicillium ochrochloron* (ATCC 9112), *Penicillium verrucosum* var. *cyclopium* (food isolate)	Heleno et al. [140]
Rare Earth-Carboxymethylated Ganoderma applanatum Polysaccharide	*Valsa mali*, *Fusarium oxysporum*, *Gaeumannomyces graminis*, *Colletotrichum gloeosporioides*, *Alternaria brassicae*	Sun et al. [141]
Ganodermin	*Botrytis cinerea*, *F*. *oxysporum*, *Physalo sporapiricola*	Wang and Ng [113]
Mycelia	*Acremonium strictum*, *A*. *glaucus*, *A*. *flavus*, *A*. *fumigatus*, *A*. *nidulans*, *A*. *niger*, *A*. *terreus*, *T*. *viride*	Ćilerdžić et al. [133]
Antiviral activity		
Ganoderiol F & Ganodermanontriol	HIV 1(HIV-1 protease)	El-Mekkawy et al. [142]
Carpophores	Herpes simplex virus types 1 (HSV-1) and 2 (HSV-2), influenza A virus (Flu A), and vesicular stomatitis virus (VSV) Indiana and New Jersey strains	El-Mekkawy et al. [142]
Acidic protein-bound polysaccharide	HSV-1 and HSV-2	Eo et al. [143]
Fruiting bodies	Oral human papillomavirus (HPV)	Donatini [144]
NG	Newcastle disease virus (anti-neuraminidase)	Zhu et al. [80], Shamaki et al. [145]
Fruiting bodies	Epstein-Barr Virus	Iwatsuki et al. [146]
Mycelia	Hepatitis B virus	Li et al. [147]
Mycelia (Ganoderic acid)	Hepatitis B	Li and Wang [148]
Lanosta-7,9(11),24-trien-3-one,15;26-dihydroxy (GLTA), Ganoderic acid Y	Enterovirus 71	Zhang et al. [79]

* **NG:** data not given.

## Data Availability

Not applicable.

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
