# Peer review of "Nutritional Profile and Health Benefits of Ganoderma lucidum “Lingzhi, Reishi, or Mannentake” as Functional Foods: Current Scenario and Future Perspectives"

_foods, 2022, doi:10.3390/foods11071030_

Round 1

Reviewer 1 Report

The comments are on the attached manuscript. There are some minor corrections.

Author Response

Response to Editors and Reviewers’ Comments

IMPORTANT: Our revised manuscript was edited according to the rules of the Journal “Foods”. As requested, I activated the "Tracked Changes" option to highlight any changes in our paper.

Responses to Reviewer (1) Comments

  • The comments are on the attached manuscript. There are some minor corrections.

Responses

  • First, many thanks to Reviewer (1) for this comment and also for describing our manuscript as “it needs minor corrections”.
  • As suggested, I did all editing corrections requested. As recommended by Reviewer 4 to re-organize our manuscript, it will be so difficult to specify the line and page numbers for each editing correction.

Reviewer 2 Report

The article is interesting and I have the following comments:  - There are two many affiliations for the author. Please indicate the current one.    - The introduction part is better to appear as a text not items.   - Please explain the names interference of lucidum and lingzhi and which is the accepted name ?   - Section 2.1. Please cite relevant new references about the activity and constituents of Ganoderma  - Section 3.2. Antiviral potential, there is an experimental animal trial article published on the activity of Ganoderma lingzhi, please cite it.

Author Response

Response to Editors and Reviewers’ Comments

IMPORTANT: Our revised manuscript was edited according to the rules of the Journal “Foods”. As requested, I activated the "Tracked Changes" option to highlight any changes in our paper.

Responses to Reviewer (2) Comments

  • The article is interesting.

Responses

  • First, many thanks to Reviewer (2) for this comment and also for describing our manuscript as “the article is interesting”.
  • I have the following comments:

(1)  There are too many affiliations for the author. Please indicate the current one.

Responses

  • Thanks to Reviewer (2) for this comment.
  • I’d like to confirm that all these affiliations are related to the author and this study. Hence for respect and appreciate viewpoint I put them.

(2) The introduction part is better to appear as a text not items.

Responses

  • Thanks to Reviewer (2) for this comment.
  • As suggested, I modified the introduction to appear as text, not items. But as recommended by Reviewer 4 to re-organize our manuscript, this part is moved. Please see the modified part entitled “1.3.1. How to define functional food?”; Pages 5,6; Lines 136 to 154.

(3)  Please explain the names interference of lucidum and lingzhi and which is the accepted name?

Responses

  • Thanks to Reviewer (2) for this comment.
  • From the botanist of botanists’ and taxonomists’ viewpoint, the correct scientific name is Ganoderma lucidum, this scientific name was used in the majority of scientific publications. But “Lingzhi” is the common name in China. Please see the explanation on Pages 2 and 3.

(4)  Section 2.1. Please cite relevant new references about the activity and constituents of Ganoderma.

Responses

  • Thanks to Reviewer (2) for this comment.
  • As suggested, I added the relevant new references about the activity and constitutes of Ganoderma. Please see Pages 10,11.
  • The relevant new references are properly cited within the text and in the list of references.

(5)  Section 3.2. Antiviral potential, there is an experimental animal trial article published on the activity of Ganoderma lingzhi, please cite it.

Responses

  • Thanks to Reviewer (2) for this comment.
  • As suggested, I cited the suggested article. Please see Page 14; Line 320.

Reviewer 3 Report

Overview and general recommendation:

The submitted work provides an extensive overview of nutraceutical value of Ganoderma lucidum and the development of commercial functional food. The manuscript is well written, clear to understand, and gives a good insight into the potential of the specific bioprocess for future perspective use. It will be favorable for the scientific community as an important summary contribution.

This paper is prepared accordingly to Foods, in these terms, I recommend minor revision.

Herein my editing comments regarding draft’s quality improvement:

  1. Line 31: correct utilize to utilizing
  2. Line 47: correct to ‘state that
  3. Line 60: correct to ‘the question that comes to mind is
  4. Line 81: replace word by to in
  5. Line 82: correct to ‘considered a curative food’
  6. Line 101: correct ‘the war’ to ‘war
  7. Line 111: correct ‘is having’ to have
  8. Line 116: correct ‘by’ to furniture
  9. Line 135: correct to ‘depends on’
  10. Line 169: correct to ‘was reported as’
  11. Line 172: pls rephrase to ‘Heavy metals affect both individual reactions and complex metabolic processes once they enter the fungal cell; for example, in lucidum, heavy metal toxicity decreases in the following order’
  12. Line 193: correct to ‘and are sold’
  13. Line 222: replace ‘belong’ to ‘belong to
  14. Line 224: correct to ‘nitritionist’s and ‘β-glucan instead of glucans
  15. Line 238: correct to ‘ranges
  16. Line 286: replace ‘radicals’ to ‘free radical
  17. Line 289: correct to ‘and cosmetics’
  18. Line 290: correct to ‘contain a proteoglucan’ or ‘have proteoglucans’
  19. Line 336: replace ‘has been’ to ‘was identified’
  20. Line 340: replace to ‘outbreaks’ instead of ‘the outbreaks’
  21. Line 369: correct to ‘potentially causing
  22. Line 384: correct to ‘declared the pneumonia outbreak that appeared in Wuhan a major
  23. Line 391: erase ‘which
  24. Line 422: change ‘with’ to ‘of
  25. Line 459: correct to ‘dysdiodsis in
  26. Line 461: correct to ‘gut microbiota’s
  27. Line 465: replace ‘reduced’ to reduce
  28. Line 493: erase from, correct to ‘tonics that improve
  29. Line 506: replace ‘mice’ to mouse
  30. Line 509-510: the sentence is not clear pls rephrase appropriately
  31. Line 512: replace ‘reducing’ to inhibiting the growth of
  32. Line 517: replace ‘are’ to ‘have been
  33. Line 519: replace ‘were’ to ‘have been
  34. Line 522: correct ‘on’ to ‘about
  35. Line 523: correct to ‘and what to do
  36. Line 534: correct ‘its’ to ‘their
  37. Line 537: correct to ‘encountering
  38. Line 544: replace ‘it was’ to ‘they were
  39. Line 547: correct to ‘e.g the unique’
  40. Line 555: correct ‘mushroom’ to ‘mushrooms
  41. Line 560: correct to ‘where they are combined
  42. Line 564: correct to ‘lucidum’s
  43. Line 569: erase ‘but’,,, correct to ‘are sold’
  44. Line 598: erase ‘and tracing
  45. Line 608: correct ‘in alleviating’ to ‘to alleviate

Author Response

Response to Editors and Reviewers’ Comments

IMPORTANT: Our revised manuscript was edited according to the rules of the Journal “Foods”. As requested, I activated the "Tracked Changes" option to highlight any changes in our paper.

Responses to Reviewer (3) Comments

  • The submitted work provides an extensive overview of nutraceutical value of Ganoderma lucidum and the development of commercial functional food. The manuscript is well written, clear to understand, and gives a good insight into the potential of the specific bioprocess for future perspective use. It will be favorable for the scientific community as an important summary contribution.

Responses

  • First, many thanks to Reviewer (3) for this comment and also for describing our manuscript as “well written, clear to understand, and gives a good insight into the potential of the specific bioprocess for future perspective use. It will be favorable for the scientific community as an important summary contribution”.
  • This paper is prepared accordingly to Foods, in these terms, I recommend minor revision.

Responses

  • Thanks to Reviewer (3) for this comment.
  • Many thanks to the Reviewer (3) for describing our manuscript as “it needs minor corrections”.
  • Herein my editing comments regarding draft’s quality improvement.

Responses

  • Thanks to Reviewer (3) for this comment.
  • As suggested, I did all editing corrections requested. As recommended by Reviewer 4 to re-organize our manuscript, it will be so difficult to specify the line and page numbers for each editing correction.

Reviewer 4 Report

The author has proposed a narrative review about the Ganoderma lucidum mushroom, discussing its composition, possible uses and biotechnological and market implications. Despite the numerous citations, the author discusses the various aspects in a very discursive way and without organizing the evidence based on the type of works cited. There is no real discussion of the data obtained from the references cited and the logical organization of the topics covered is a little confused and verbose. From the very first lines, it is perceived that the manuscript will treat the evidence partially to highlight the positive characteristics of G. lucidum, overshadowing the problems related to safety and the lack of sufficient clinical trials to demonstrate the properties attributed to the mushroom. The result is a biased discussion with hyperbolic sentences and references used to theses of the suggested hypotheses, even if not always sufficient.

We recommend a reorganization of the manuscript with a more objective approach to the evidence and which shows the literature data more clearly, highlighting when the evidence is based on in vitro studies, studies on animal models or clinical studies on humans.

Specifically, I suggest focusing attention on some passages, hoping that they will be useful for the author to implement the manuscript:

- In the abstract, the author refers in particular to prospective studies. Does he mean observational studies or does he want to express another concept?

- The introduction seems to be focused on functional foods and intestinal health and diseases such as chronic intestinal inflammatory diseases and celiac disease, for which there is a multifactorial aetiology. It should be reorganized to propose the background about the topic (G. lucidum), removing the superfluous and unscientific

- Even in the discussion on functional foods, there is no exhaustive discussion: what is the approach to the concept of functional food in the various countries? Can a functional food also be an extract?

- Paragraph 1.3.3 talks about G. lucidum's ability in bioremediation. However, this raises the hypothesis that these mushrooms may contain high concentrations of pollutants. The author should deepen this aspect.

- The data used for Figure 3 should be discussed in more detail. Does the graph refer only to publications on Elsevier? Why hasn't the data been pulled from a larger container like PubMed?

- Chapter 2 I believe is the weakest. The author discusses nutritional aspects considering G. lucidum as a functional food, however, he admits that it is not used as a food but as a medicinal mushroom. Furthermore, especially in the case of table 2 and the descriptive text, there are numerous inconsistencies: it is not clear what the protein concentration is, whether the one indicated in lines 231-232, those in the table or the one in 238. In table 2 it is not very clear what the author means by water-soluble proteins: percentage of total proteins present or percentage by weight? In the second case, it seems unlikely. Even the statement in lines 241-243 makes no sense! Assuming that there are those protein concentrations (35-42g / 100g) it would be impossible to consume 100g of dried mushrooms. The author must have confused the literature data. I recommend reviewing this part or removing it (considering he's not talking about edible mushrooms). Furthermore, the statements done by the author do not emerge from references 69, 71, 71. In the first place, the cited manuscripts do not analyze G.lucidum about the protein content and it is never stated that the proteins of mushrooms have digestibility comparable to that of dairy products. If anything, it is claimed that they are comparable to other vegetable proteins but lower than those of meat (notoriously less digestible than milk proteins). Also, to reach certain quantities, the indigestible carbohydrate content would be prohibitive. This part, in particular, is very speculative. Likewise, it is unthinkable to consider G. lucidum as a nutritional source to prevent malnutrition in developing countries (lines 255-257).

- Table 2 should show the mean contents and the standard deviation for both dry and wet weight, eliminating the references to DRIs which, as explained, have no meaning in this context. Similarly, the value in Kcal should be clarified, because it seems unlikely (it is advisable to check and specify between dry weight and fresh weight, both mean and standard deviation).

- Regarding the topics on health benefits, it would be better to create a table with a short description of the studies mentioned, specifying the type of study (in vitro, in vivo, clinical study on humans), the study design (observational, intervention, etc.), the number of individuals involved, the presence of control or placebo, the main outcomes and possibly the statistical significance of data

- Table 4 does not show sufficient data

- The paragraphs on antiviral effects are very speculative. The paragraphs are dedicated to the description of the single viral agents with a minimal part dedicated to the possible roles of G. lucidum

- Lines 487-498 show a digression that has little to do with the topic of the paragraph

- In the conclusions, the concept of limited intervention studies on humans and especially limited studies concerning the safety of use should be stressed. The same foresight should be included in the infographic if you want to maintain objectivity

- Many statements, also included in the infographic, should be mitigated with the use of conditional verbs

- On line 620 “time has now come” is too hyperbolic. Even from reading the manuscript alone, I could say the exact opposite. I recommend greater caution

Minor aspects:

- In line 62, analyzes mean converts?

- In line 80 does prohibit mean prevent? Also, add a conditional verb

- In line 93 it would be better to replace the word "enormous", which is too hyperbolic

- On lines 145 and 233 there are formatting errors for references

- In line 160 it is better to replace the positive term with improving

- In the tables, the numbering of the references does not respect progressivity

- In line 281 type of mushroom means components?

- Lines 313-315 seem out of place if the author is talking about proteins

- In line 292 it would be better to replace the word " incomparable ", too hyperbolic

- There is excessive use of bold and italics not necessary (for example, lines 573 and 584

Author Response

Response to Editors and Reviewers’ Comments

IMPORTANT: Our revised manuscript was edited according to the rules of the Journal “Foods”. As requested, I activated the "Tracked Changes" option to highlight any changes in our paper.

Responses to Reviewer (4) Comments

  • The author has proposed a narrative review about the Ganoderma lucidum mushroom, discussing its composition, possible uses and biotechnological and market implications. Despite the numerous citations, the author discusses the various aspects in a very discursive way and without organizing the evidence based on the type of works cited. There is no real discussion of the data obtained from the references cited and the logical organization of the topics covered is a little confused and verbose. From the very first lines, it is perceived that the manuscript will treat the evidence partially to highlight the positive characteristics of G. lucidum, overshadowing the problems related to safety and the lack of sufficient clinical trials to demonstrate the properties attributed to the mushroom. The result is a biased discussion with hyperbolic sentences and references used to theses of the suggested hypotheses, even if not always sufficient.

Responses

  • Thanks to Reviewer (4) for this comment.
  • As suggested, I modified the whole manuscript accordingly.
  • The relevant new references are properly cited within the text and in the list of references.
  • We recommend a reorganization of the manuscript with a more objective approach to the evidence and which shows the literature data more clearly, highlighting when the evidence is based on in vitro studies, studies on animal models or clinical studies on humans.

Responses

  • Thanks to Reviewer (4) for this comment.
  • As recommended, I re-organized the manuscript to be a more objective approach to the evidence and which shows the literature data more clearly, highlighting when the evidence is based on in vitro studies, studies on animal models, or clinical studies on humans if the data are available.
  • The relevant new references are properly cited within the text and in the list of references.
  • Specifically, I suggest focusing attention on some passages, hoping that they will be useful for the author to implement the manuscript:

(1)  In the abstract, the author refers in particular to prospective studies. Does he mean observational studies or does he want to express another concept?

Responses

  • Thanks to Reviewer (4) for this comment.
  • As suggested, I modified the abstract section. Please see Page 1; Lines 27 to 32.

(2)  The introduction seems to be focused on functional foods and intestinal health and diseases such as chronic intestinal inflammatory diseases and celiac disease, for which there is a multifactorial etiology. It should be reorganized to propose the background about the topic (G. lucidum), removing the superfluous and unscientific.

Responses

  • Thanks to Reviewer (4) for this comment.
  • As suggested, I re-organized the “Introduction Section” and removed the superfluous and unscientific. Please see Pages 1 to 7; Lines 36 to 209.
  • The relevant new references are properly cited within the text and in the list of references.

(3)  Even in the discussion on functional foods, there is no exhaustive discussion: what is the approach to the concept of functional food in the various countries? Can a functional food also be an extract?

Responses

  • Thanks to Reviewer (4) for this comment.
  • As suggested, I added a new subsection entitled “1.3.2. What do the definitions of functional foods conclude?” to give the answers for the 2 questions. Please see Page 6; Lines 160 to 166.
  • The relevant new references are properly cited within the text and in the list of references.

(4)  Paragraph 1.3.3 talks about G. lucidum's ability in bioremediation. However, this raises the hypothesis that these mushrooms may contain high concentrations of pollutants. The author should deepen this aspect.

Responses

  • Thanks to Reviewer (4) for this comment.
  • I agreed with Reviewer 4. Although, Ganoderma lucidum can provide a promising mechanism for heavy metal uptake and accumulation (Ipeaiyeda et al. 2020), and also it is known to possess excellent tolerance for heavy metals (Gupta et al., 2019). But with the increased interest in G. lucidum as a promising source of several bioactive compounds (which have nutritional and medicinal effects) that are present in all parts of the fungus (fruit bodies, mycelium, and spores). Therefore, the use of G. lucidum to decontaminate polluted environments is not expedient because of its excellent medicinal properties (Ipeaiyeda et al. 2020). For that, I deleted this paragraph because of its unsuitability with our manuscript topic.

(5)  The data used for Figure 3 should be discussed in more detail. Does the graph refer only to publications on Elsevier? Why hasn't the data been pulled from a larger container like PubMed?

Responses

  • Thanks to Reviewer (4) for this comment.
  • I’d like to mention that our manuscript is “Normal Review Article” NOT “Systematic Review”. So, I think this part is enough and don’t need more details which the same style was published before, for example:

- El Sheikha AF*, Ray RC.  (2022). Bioprocessing of horticultural wastes by solid-State fermentation into value-added/innovative Bioproducts: A review. Food Reviews International. (In press). DOI: 10.1080/87559129.2021.2004161.

  • The graph referred to publications on Scopus, and I’ll give your suggestion the full consideration in the future work to pull the data from a larger container like PubMed.

(6)  Chapter 2 I believe is the weakest. The author discusses nutritional aspects considering G. lucidum as a functional food, however, he admits that it is not used as a food but as a medicinal mushroom. Furthermore, especially in the case of table 2 and the descriptive text, there are numerous inconsistencies: it is not clear what the protein concentration is, whether the one indicated in lines 231-232, those in the table or the one in 238. In table 2 it is not very clear what the author means by water soluble proteins: percentage of total proteins present or percentage by weight? In the second case, it seems unlikely. Even the statement in lines 241-243 makes no sense! Assuming that there are those protein concentrations (35-42g / 100g) it would be impossible to consume 100g of dried mushrooms. The author must have confused the literature data. I recommend reviewing this part or removing it (considering he's not talking about edible mushrooms). Furthermore, the statements done by the author do not emerge from references 69, 71, 71. In the first place, the cited manuscripts do not analyze G.lucidum about the protein content and it is never stated that the proteins of mushrooms have digestibility comparable to that of dairy products. If anything, it is claimed that they are comparable to other vegetable proteins but lower than those of meat (notoriously less digestible than milk proteins). Also, to reach certain quantities, the indigestible carbohydrate content would be prohibitive. This part, in particular, is very speculative. Likewise, it is unthinkable to consider G. lucidum as a nutritional source to prevent malnutrition in developing countries (lines 255-257).

Responses

  • Thanks to Reviewer (4) for this comment.
  • As suggested, many changes have been done in Chapter 2.
  • Despite Ganoderma lucidum is not edible in its raw state due to its higher content of bitter compounds, its palatability will be increased by turning it into manufactured products such as powders, supplements, and tea (Bryant et al., 2017).
  1. G. lucidum is used as a supplement, so it is very important to show its nutritional value and the coverage of dietary recommended intake.
  • The relevant new references are properly cited within the text and in the list of references.
  • Please see Pages 7 to 12; Lines 206 to 298.

(7)  Table 2 should show the mean contents and the standard deviation for both dry and wet weight, eliminating the references to DRIs which, as explained, have no meaning in this context. Similarly, the value in Kcal should be clarified, because it seems unlikely (it is advisable to check and specify between dry weight and fresh weight, both mean and standard deviation).

Responses

  • Thanks to Reviewer (4) for this comment.
  • I’d like to mention that the data in Table 2 were cited according to the original sources which didn’t use the standard deviation. Hence, I must respect the original data.
  1. G. lucidum is used as a supplement, so it is very important to show its nutritional value and the coverage of dietary recommended intake.
  • As confirmed by all Reviewers 1, 2, and 3. The table is clear.
  • All references of the original data were cited properly.

(8)  Regarding the topics on health benefits, it would be better to create a table with a short description of the studies mentioned, specifying the type of study (in vitro, in vivo, clinical study on humans), the study design (observational, intervention, etc.), the number of individuals involved, the presence of control or placebo, the main outcomes and possibly the statistical significance of data.

Responses

  • Thanks to Reviewer (4) for this comment.
  • I’d like to mention that the majority of data needed to build the requested table is not available. Therefore, as confirmed by all Reviewers 1, 2, and 3, the current style is adequate.

(9) Table 4 does not show sufficient data.

Responses

  • Thanks to Reviewer (4) for this comment.
  • As confirmed by all Reviewers 1, 2, and 3. Table 4 shows sufficient data.

(10) The paragraphs on antiviral effects are very speculative. The paragraphs are dedicated to the description of the single viral agents with a minimal part dedicated to the possible roles of G. lucidum.

Responses

  • Thanks to Reviewer (4) for this comment.
  • As suggested, I supported this part with experimental studies on the animal.
  • The relevant new references are properly cited within the text and in the list of references.

(11) Lines 487-498 show a digression that has little to do with the topic of the paragraph.

Responses

  • Thanks to Reviewer (4) for this comment.
  • As suggested, I deleted the paragraph and related references. Please see Page 18; Line 503.

(12) In the conclusions, the concept of limited intervention studies on humans and especially limited studies concerning the safety of use should be stressed. The same foresight should be included in the infographic if you want to maintain objectivity

Responses

  • Thanks to Reviewer (4) for this comment.
  • As suggested, I modified the conclusions. Please see Page 30; Lines 629 to 633.

(13) Many statements, also included in the infographic, should be mitigated with the use of conditional verbs.

Responses

  • Thanks to Reviewer (4) for this comment.
  • As suggested, I modified the text of the infographic. Please see Page 21.

(14) On line 620 “time has now come” is too hyperbolic. Even from reading the manuscript alone, I could say the exact opposite. I recommend greater caution.

Responses

  • Thanks to Reviewer (4) for this comment.
  • As suggested, I modified this paragraph. Please see Page 30; Lines 629 to 633.
  • Minor aspects.

Responses

  • Thanks to Reviewer (4) for this comment.
  • As suggested, I did all editing corrections requested. As recommended to re-organize our manuscript, it will be so difficult to specify the line and page number for each editing correction.
  • In the tables, I re-numbered the references in respect of progressivity.

Round 2

Reviewer 4 Report

even if in part, the author has answered my doubts. There was a lot of work on revising the manuscript. Unfortunately, I think the manuscript has poor scientific relevance. 

It is very annoying to read that my suggestions are ignored because the other referees have not highlighted these issues. A table with insufficient data, even if reported by another article as it is, does not make it adequate.  Furthermore, the fact that the paper is not a systematic review does not imply that it must be approximative.